# An Epsilon-Frontier for Faster Optimization in Nonlinear Manifold Learning

## Arthur R. Drake, Qiuyi Chen, and Mark D. Fuge

Informatics for Design, Engineering And Learning (IDEAL) Lab
Department of Mechanical Engineering
University of Maryland, College Park, MD, USA
adrake17@umd.edu, qchen88@umd.edu, fuge@umd.edu

## Abstract

Complex engineering problems such as compressor blade optimization often require large amounts of data and computational resources to produce optimal designs because traditional approaches only operate in the original high-dimensional design space. To mitigate this issue, we develop a simple yet effective autoencoder architecture that operates on a prior $\epsilon$-frontier from examples of past optimization trajectories. This paper focuses on using such nonlinear methods to maximize dimensionality reduction on an easily verifiable synthetic dataset, providing a faster alternative to high-fidelity simulation techniques. We test a variety of component reduction models on the $\epsilon$-frontier of a synthetic 2-dimensional dataset of $K$ trajectories, for which we can easily verify the accuracy of alterations to the latent space. We find that our autoencoder generally converges more quickly than other simple architectures such as PCA in the resulting 1-dimensional space.

## Introduction

In order to address the problem of growing complexity in problems within engineering and other fields, it is helpful to produce a new low-dimensional representation of some subset of the original data. One important reason for this is that the amount of data required to find an optimum increases exponentially with the number of dimensions (Bittner 1962), part of the so-called curse of dimensionality. With this in mind, the resulting low-dimensional space should also exhibit sufficient continuity and accuracy so that it can be easily explored, producing consistent outputs when converted back to the original space. For example, this may assist in predicting design variables that would otherwise need to be calculated by multi-physics simulators, which often take orders of magnitude longer to run as compared to data-based approaches. Popular categories of methods to address the high dimensionality of data have included vector quantization, various forms of Principal Component Analysis (PCA), and Generative Topological Mapping (GTM), among others (Sorzano, Vargas, and Montano 2014).

Building on these techniques, Manifold Learning aims to find a continuous distribution along a low-dimensional space, or manifold, which retains nearly all characteristics of the original data. This allows for much quicker processing and training with a minimal loss in overall design variance. However, most existing algorithms assume that examples on this manifold are readily available, which is often not the case in real-world problems. Our work addresses this issue by proposing a method of incorporating prior optimization runs to learn a low dimensional representation. We explore this method's effectiveness for simple and easily replicable test models derived from synthetic data. This paper contributes the following:

1. We first present a new algorithm that leverages an $\epsilon$-frontier envelope on a database of $K$ trajectories to learn a low-dimensional subspace wherein an optimal subset of points may lie, with the benefit of a continuous set of near-optimal points along the resulting manifold.

2. We demonstrate the effect of $\epsilon$ on the convergence rates of a Bayesian optimization algorithm, in terms of finding a new optimal point along the manifold, for two synthetic test functions. We measure this via a decrease the number of iterations needed to reach equivalent performance in the Bayesian optimizer. This directly compares the properties of each low-dimensional representation produced by the corresponding test model.

3. We show that a nonlinear autoencoder is capable of faster convergence and better overall performance compared to linear PCA for most cases, as well as KPCA when the input frontier features a discontinuity.

## Related Work

Traditionally, manifold learning algorithms have relied on constructing a nearest neighbor graph on a given set of input points, linearly approximating the local manifold geometry for each point, and minimizing a global error function to obtain the overall embedding by solving an eigenvalue problem (Zhang, Wang, and Zha 2012). An analysis by Anowar, Sadaoui, and Selim (2021) compared several popular manifold-based dimensionality reduction algorithms including ISOMAP, LLE, and t-SNE, all of which are nonlinear and unsupervised. It found that such methods generally outperform random projection-based feature extraction methods, despite some present flaws such as topological instability in ISOMAP. Several augmentations of these base-

line methods have been researched to address topics such as the adaptive selection of neighborhood sizes and accurate fitting of local geometric structures, with fairly promising results. Gu et al. (2017) devised a method utilizing the law of cosines between each data point and a principal connection curve to create the lower-dimensional geometries. This yielded improved performance in several pattern recognition applications with high-dimensional input data. Meanwhile, Wang et al. (2014) developed a generalized autoencoder with similar architecture to the one used in this paper, which was also used to reconstruct a set of new points rather than the existing ones. They further propose a deep version of the model with additional hidden layers, and combine this with Linear Discriminant Analysis (LDA) to produce highly distinct clusters of digits from the MNIST dataset. Furthermore, some attempts have been made to apply a Pareto-optimal set (PS) in the area of manifold learning. Li and Kwong (2014) proposed a general framework for evolutionary multi-objective optimization that uses the Laplacian maps algorithm to find a manifold representation of a predetermined PS of points. They noted the regularity property of an $m-1$ dimensional PS, where $m$ is the number of objectives, allowing for a smooth manifold approximation along the PS.

This paper proposes a more broadly applicable approach to Pareto-based manifold learning: it uses several test models to encode a select portion the original test data determined by an $\epsilon$-frontier, learn the resulting low-dimensional space to construct a manifold, and finally traverse that manifold to provide a set of target outputs when decoded back to the design space. This yields a generalized method to predict new optimal points along a given Pareto front.

## Methods

The following section describes our approach to reducing the complexity of a synthetic dataset. Our general approach is as follows: we first train a model (which must be capable of encoding to and decoding from a latent space) on the distribution of a previously determined $\epsilon$-frontier to create a low-dimensional manifold. This provides the advantage of near-optimality for any points chosen close to the frontier, which in this case lie on the continuous manifold. We choose arbitrary points in this space and, when decoded, determine how well they line up with the original distribution. Each iteration of this allows for an intermediate optimizer to converge towards a new desired optimum (which was not present in the original dataset) along the manifold.

### Creation of $\epsilon$-Frontiers from a Synthetic Trajectory Database

We design a 2D toy optimization problem of minimizing a loss function $\mathcal{L}(\mathbf{x} \mid t)$ to get $\mathbf{x}^\star(t)$, which simulates an optimal point in some design space, subject to a changing input condition $t$. The loss function is constructed by a Gaussian-mixture-like model over $t \in [0, 1]$ as plotted in Figure 1, for which we first design two 1D manifolds in $\mathbb{R}^2$:

$$\mathbf{x}_1(t) = \begin{bmatrix} t \\ \frac{2}{3}\{10(t-0.7)^3 + 5(t-0.7)^2 + 0.1(t-0.7) + 0.5\} \end{bmatrix} \tag{1}$$

$$\mathbf{x}_2(t) = \mathbf{x}_1(t) + \begin{bmatrix} -0.5 \\ 0.5 \end{bmatrix} \tag{2}$$

and two weight functions

$$w_1(t) = t, \quad w_2(t) = 1 - t \tag{3}$$

after which we construct $\mathcal{L}(\mathbf{x} \mid t)$ via

$$\mathcal{L}(\mathbf{x} \mid t) = w_1(t) \cdot g(\mathbf{x} \mid \mathbf{x}_1(t)) + w_2(t) \cdot g(\mathbf{x} \mid \mathbf{x}_2(t)) \tag{4}$$

where

$$g(\mathbf{x} \mid \mathbf{y}) = 1 - \exp(-20\|\mathbf{x} - \mathbf{y}\|_2^2) \tag{5}$$

To generate trajectories, the loss function $\mathcal{L}(\mathbf{x} \mid t)$ is then optimized via Bayesian optimization with upper confidence bound as the acquisition function and five initialization points for the Gaussian process prior. The optimization histories or trajectories are then stored and analyzed as per below. For the "hard" trajectory dataset, we define the offset as shown in $\mathbf{x}_2(t)$ above, and for the "easy" dataset, we set the offset to the zero vector, such that there is effectively a single Gaussian. As one example, the trajectories of different optimization methods for $\mathcal{L}(\mathbf{x} \mid t = 0.5)$ are shown in Figure 2.

The synthetic data described above was sampled from a set of K trajectories with a specified $\epsilon$ value from 0 to 1, corresponding to the quantile of most optimal values extracted from each trajectory. Also specified was the number of input $t$ points, of which there were 50 per trajectory. The sparsity of input data was further controlled by the number of trajectories included at each $t$ point, ranging from 1 to 10. Three test models were then assigned a certain latent space width, which was set to 1 in this case given the input was only comprised of two dimensions.

The models tested in this experiment were Principal Component Analysis (PCA) and a kernel-based version referred to as KPCA, both from the Python Scikit-learn library (Pedregosa et al. 2011), as well as a nonlinear autoencoder (AE) created in PyTorch (Paszke et al. 2019). This feed-forward AE features a symmetric architecture that first passes the input through two 8-dimensional linear hidden layers, each followed by an ELU activation function. This leads to the central latent space of interest, which has one dimension in the synthetic data case. The decoder simply completes the reverse of this process to translate the latent space back to two dimensions.

## Experimental Process and Results

In this section, we detail our exact process to test and visualize the convergence rates for each synthetic data model. We provide our reasoning for using various test parameters and explain the overall significance of the resulting convergence plots. Future work on this problem will also apply the autoencoder within the context of a complex multi-fidelity model, which will further validate its ability to produce accurate and smooth manifolds.

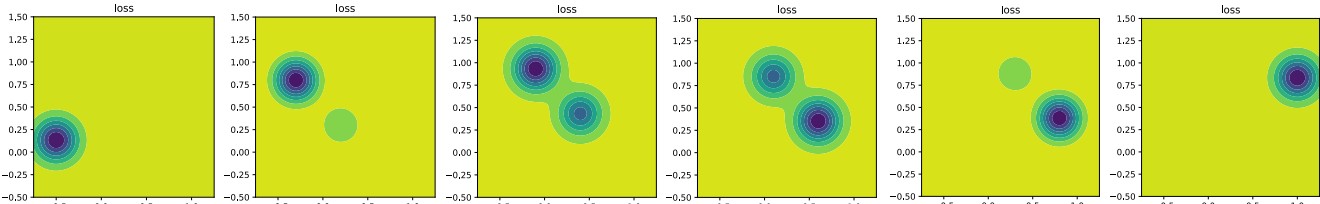

Figure 1: $\mathcal{L}(\mathbf{x} \mid t)$ at $t = 0.0,\ 0.2,\ 0.4,\ 0.6,\ 0.8,\ 1.0$ from left to right.

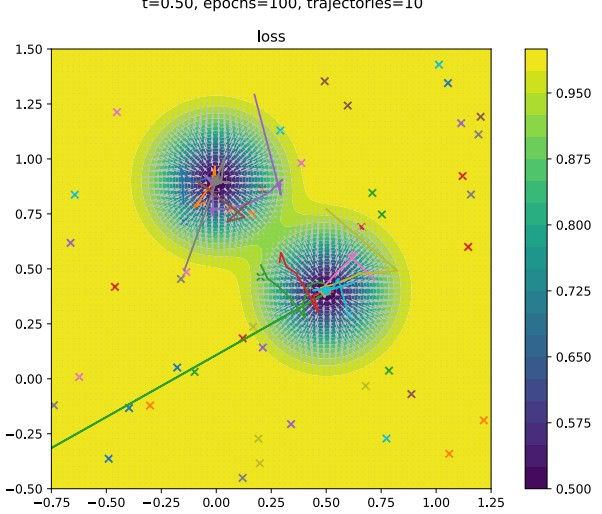

Figure 2: Example Optimization trajectories of Bayesian optimization on $\mathcal{L}(\mathbf{x} \mid t = 0.5)$.

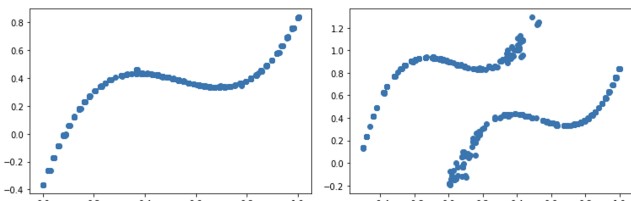

Figure 3: The easy (left) and hard (right) synthetic datasets. At $t = 0.5$, the hard dataset jumps from the left curve to the right curve as the two modes of the Gaussian Mixture Model shift.

## Convergence Rates of Bayesian Optimization on the Synthetic Frontier Manifolds

For each experiment run, a certain combination of $\epsilon$ quantile, model type, number of t points, and number of trajectories per t point was assessed. A frontier representing the $\epsilon$ quantile of most optimal points was created, which the specified model was trained on in order to create a manifold. In the case of KPCA, we used the RBF kernel, as well as tuning the hyperparameters $\alpha$ (regularization strength) and $\gamma$ (kernel bandwidth) via Cross Validation to better capture the frontier characteristics. The AE was trained within the architec-

ture designed above, using an LBFGS optimizer. Combinations of experiment parameters were tested for the following ranges:

- $\epsilon = 0.25, 0.5, 1.0$
- Model = PCA, KPCA, AE, 2D PCA
- Number of t points = 4, 6, 8, 10 for easy data; 10, 15, 20, 25 for hard data
- Number of Trajectories per t point = 1, 2

Note that the test was run with more t points for the hard data, since a lower amount resulted in training data that was too inconsistent to fit each model adequately. This is due to the discontinuity which occurs at $t = 0.5$, which as a result requires more examples to capture the two separate curves. Once the above factors were determined for a given run, we calculated the manifold search bounds for use in the Bayesian Optimization. The resulting optimization convergence test measured training loss after 30 epochs, and began with a set of 5 randomly selected points on the manifold. As mentioned previously, we constructed a Gaussian Process with re-optimized kernel parameters on each iteration (with respect to the MLL), sampling new points according to an Upper Confidence Bound algorithm. Therefore the ability of the given model to create a consistent, accurate manifold from the frontier data was directly tested.

Figure 4 shows the convergence rates of each tested model type for the easy data. 2D PCA was consistent in reaching the lowest training error, which is justified by the lack of reconstruction loss given its 2-dimensional latent space. As a result, it is reasonable to conclude this model will always converge to the optimum.

As the data becomes less sparse with increasing t points and number of trajectories, the nonlinear 1-dimensional models begin to converge to low loss values as well. In fact, they converge at a much quicker rate than 2D PCA for $N_t = 10$, $N_{traj} = 2$. This can be explained by the benefit of additional data points for the nonlinear models, as they adapt to the increasingly clear 1D manifold. In turn, the Bayesian Optimization only has to optimize along one dimension rather than two, providing an acceleration. We also note that the required amount of t points and trajectories for rapid convergence is still small in comparison to the size of the entire dataset, showing that the creation of an $\epsilon$ frontier as a preprocessing tool is greatly helpful to reduce data volume. Finally, 1D PCA is limited by its linear nature; it can never converge to the true nonlinear optimum in this case.

A separate optimization gap analysis was also conducted

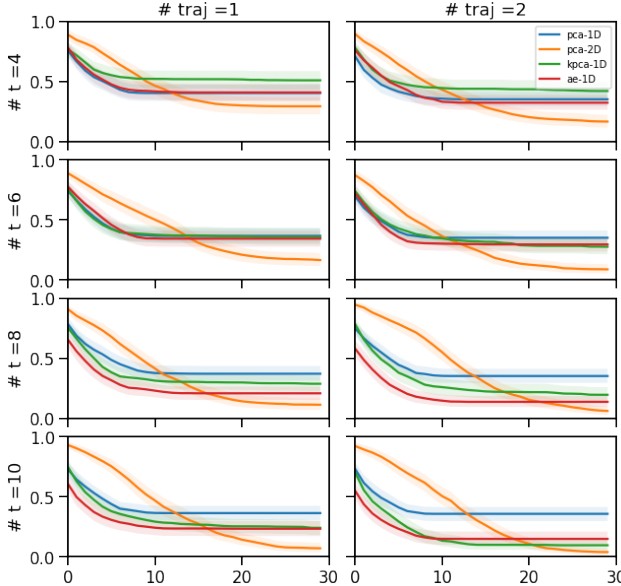

Figure 4: The convergence results for the four models tested on the easy synthetic data across 30 epochs.

for each $\epsilon$ value tested. This allowed us to directly compare the effect of including more suboptimal points found at higher $\epsilon$. We found that $\epsilon = 0.25$ and $\epsilon = 0.5$ performed at roughly equal levels, while $\epsilon = 1.0$ exhibited more error in addition to its heightened training time. Therefore, we chose $\epsilon = 0.25$ for continued testing and convergence visualization.

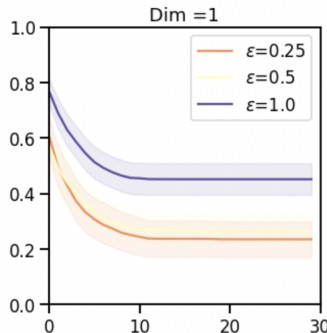

Figure 5: The autoencoder optimization gap at each $\epsilon$.

When shifting to the discontinuous hard data, it immediately becomes evident that none of the 1D models are capable of converging very well, even with a large amount of t points. We note that 2D PCA, which is our control model, still converges because it does not use any actual dimensionality reduction. The discontinuity sufficiently disrupts the manifold to the point that a 1D representation is simply not sufficient for these models. However, the autoencoder visibly separates from KPCA and PCA given more input examples with one trajectory, indicating that it performs significantly better over a discontinuous frontier. Overall, how-

ever, a much more flexible architecture is needed to address the discontinuity. For example, Khayatkhoei, Elgammal, and Singh (2019) propose a multi-generator GAN method which learns a prior over the generators rather than using a fixed prior. We will explore the use of such advanced models in future research.

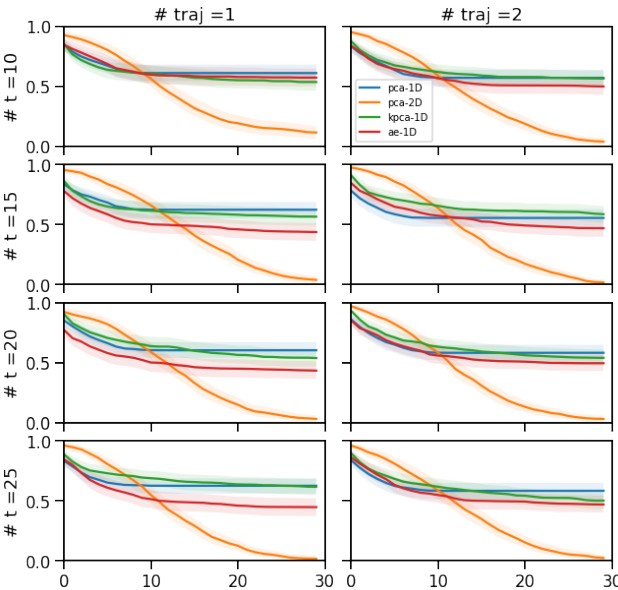

Figure 6: The convergence results for the four models tested on the hard synthetic data across 30 epochs.

## Conclusion

This paper explores the performance of linear and nonlinear models over the low-dimensional manifolds of an example problem's optimization results using various test cases and parameters. The reduction of the design space to a manifold allows for significantly quicker model convergence with minimal loss of variance in the design context. We showed that a fairly smooth and continuous manifold may be traversed by a simple model such as an autoencoder, producing qualitatively successful results when translated back into the design space. We also demonstrated the impact of data sparsity on the tested models, showing that using even a relatively small segment of the original samples can yield rapid convergence on the resulting manifold. Future work will include model design for discontinuous and other poorly-behaving manifolds, as well as an expanded study of applications in high-dimensional problems. There will also be a greater emphasis on both empirical and theoretical analysis of manifold-based model results in the decoded design space as compared to the outputs of physics-based models.

## Acknowledgements

This research was supported in part by funding from the U.S. Department of Energy's Advanced Research Projects Agency-Energy (ARPA-E) DIFFERENTIATE funding opportunity through award DE-AR0001201.

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
