# OpenReview forum: "An Epsilon-Frontier for Faster Optimization in Nonlinear Manifold Learning"
_AAAI.org/2022/Workshop/ADAM — AAAI 2022 Workshop ADAM_

### Official Review · Reviewer_nfk4 · 2021-12-01
**Interesting work that could benefit from better presentation.**

**Rating:** 7
**Confidence:** 3

**Review:**

The paper introduces the epsilon-frontier algorithm coupled with an autoencoder to accelerate the optimization in high-dimensional spaces. This interesting approach chooses only a subset of data (close to the current optimum). It is timely work with high potential.

The paper proposes the generic approach but only shows the results for the toy example. Moreover, most plots are challenging to decipher; some are not referenced in the paper (figure 5) and axis labels (reader needs to read the document to find the meaning of axis carefully). How generic is this observation from Figure 5? Was epsilon selected once? I wonder if some adaptive strategy could help to make this method less dependent on epsilon.

---

### Official Review · Reviewer_Zot6 · 2021-12-01
**Nice paper combining manifold learning and Bayesian optimization**

**Rating:** 7
**Confidence:** 4

**Review:**

The paper introduces a new algorithm that combines elements of Bayesian optimization, nonlinear manifold learning, and autoencoder training to accelerate optimization in high-dimensional design spaces. The algorithm is validated on two toy datasets, and several properties of convergence (both in terms of training time as well as number of samples) are reported

The paper is nicely written and the direction seems compelling. Some (minor) points of feedback:
- Consider validating on more challenging/realistic scenarios.
- Label both axes in figs 3,4,5,6.